# Understanding Root Rot Disease in Agricultural Crops

**Bruce A. Williamson-Benavides** [1] **and Amit Dhingra** [1,2,]*

1   Molecular Plant Sciences Program, Washington State University, Pullman, WA 99164, USA; b.williamsonbeanvid@wsu.com
2   Department of Horticulture, Washington State University, Pullman, WA 99164, USA
*   Correspondence: adhingra@wsu.com

**Abstract:** Root rot diseases remain a major global threat to the productivity of agricultural crops. They are usually caused by more than one type of pathogen and are thus often referred to as a root rot complex. Fungal and oomycete species are the predominant participants in the complex, while bacteria and viruses are also known to cause root rot. Incorporating genetic resistance in cultivated crops is considered the most efficient and sustainable solution to counter root rot, however, resistance is often quantitative in nature. Several genetics studies in various crops have identified the quantitative trait loci associated with resistance. With access to whole genome sequences, the identity of the genes within the reported loci is becoming available. Several of the identified genes have been implicated in pathogen responses. However, it is becoming apparent that at the molecular level, each pathogen engages a unique set of proteins to either infest the host successfully or be defeated or contained in attempting so. In this review, a comprehensive summary of the genes and the potential mechanisms underlying resistance or susceptibility against the most investigated root rots of important agricultural crops is presented.

**Keywords:** fungi; oomycetes; resistance; susceptibility; molecular mechanisms; quantitative trait loci; mapping





## 1. Introduction

Root rots have a significant impact on global crop production [1]. Depending on the causal agent, host susceptibility, and the environmental conditions, crop losses can range from slightly above the economic threshold to losing complete fields [2–4]. Interestingly, legumes seem to be the most common host for these pathogens [3,5,6]. However, monocots and dicots, cereals and legumes, fruit trees, and tubers also fall prey to root rots.

Fungi and oomycetes most commonly cause root rot disease. However, bacteria and even viruses can be the causal agents [4,7–12]. Due to more than one pathogen's involvement, the disease is commonly referred to as a root rot complex. Some classic examples include the black root rot of strawberry attributed to Pythium (oomycete), Fusarium (fungus), and Rhizoctonia (fungus) pathogens [13–15], and the pea root rot complex caused by *A. euteiches* (oomycete), *F. oxysporum*, *F. solani*, *F. avenaceum*, *Mycosphaerella pinodes* (fungus), Pythium spp., *R. solani*, and Phytophthora spp. (oomycete) [16–19].

Unless the root rot complex affects seed germination, the root-specific symptoms go unnoticed or are not visible. If symptoms appear aboveground, the plants usually fail to recover. Some of the symptoms associated with root rots include browning and softening of root tips, root lesions that vary in size and color (reddish, brown, and black), yellowing and wilting of leaves, stunted plant growth, reduced yield, and loss of crop [1,3,4,20–22]. Selected root rot pathogens can also cause post-harvest rots in beets, potato, and sweet potato. The proliferation of root rot pathogens is favored by moderate to high soil moisture, poor drainage conditions, soil compaction, the optimal temperature for pathogen growth, mono-cropping, and other factors that contribute to plant stress [1,23–25]. The unpredictable climatic conditions portend an increase in mean temperatures and other

natural calamities such as droughts, floods, and storms. These conditions are expected to inflict constant stress on crops, which is expected to favor the increased activity of root rot pathogens [26–28].

Cultural, physical, biological, and chemical control methods have been used as management strategies to control root rot disease. However, to date, these strategies have only been partially successful. Most of the root rot pathogens are distributed globally, and some species can survive up to 10 years in the soil [29]. Several root rot pathogens are host-specific, however, some have a wide range of hosts. Therefore, crop rotation may not be fully effective as a control method [3,29]. Chemical control is often inefficient due to these pathogens' soilborne nature and is not the most sustainable option as it also impacts beneficial microbes. Furthermore, there is high likelihood of cross contamination between contiguous plots and when using shared field equipment [30,31].

There is a critical need to understand the genetic basis of root rots and incorporate the information in breeding strategies to develop root rot-resistant crops. The current understanding of plant molecular defense responses is derived primarily from studies using foliar pathosystems [32]. Specific and unique genetic and molecular aspects of the host-pathogen interactions in the roots have been unraveled in the past few decades. This review summarizes the different groups of root rot species that affect crops, instances of host resistance and susceptibility, and the genes and proposed molecular mechanisms underlying host-pathogen interactions.

## 2. Common Causal Agents of Root Rots

### 2.1. Fungi

Fungi represent one of the most predominant root rot causing agents. The most studied and problematic fungal root rots are the Rhizoctonia root rot, Fusarium root rot, Phoma root rot, and Black root rot. They account for incalculable yield losses across agricultural and horticultural crops. These fungal pathogens also impact the wood industry.

Rhizoctonia root rot is caused by the soilborne fungus *Rhizoctonia solani* (Table 1). *R. solani* is a species complex because of the many related but genetically distant isolates. Isolates are classified into 12 anastomosis groups (AG) based on their hyphal incompatibility and their host specificity (Table 1) [33]. For instance, AG2-1 and AG4 are associated with stem and root rot diseases in dicots such as Brassicaceae species, while AG8 causes root rot in monocots [22,34]. In general, the first four AGs (AG-1, -2, -3, and -4) cause important diseases in plants worldwide, whereas the remaining AGs (AG-5, -6, -7, -8, -9, -10, -11, -12) are less destructive and generally have a restricted geographic distribution (Table 1) [21].

Symptoms of *R. solani* vary among species, but it primarily affects underground tissues (seeds, hypocotyls, and roots); however, it can also infect above-ground plant parts such as pods, fruits, leaves, and stems. Pre- and post-emergence damping off is the most common symptom of *R. solani*. Surviving seedlings can often develop reddish-brown lesions (cankers) on stems and roots. This pathogen can occasionally infect fruit and leaf tissue near or on the soil surface [21,22,35]. *R. solani* is responsible for high yield losses in many crops, and some of the noteworthy examples are highlighted in Table 1.

**Table 1.** Crop species affected by *Rhizoctonia solani* root rot.

| Crops spp. | Disease Name | AG | Resistance | References |
|---|---|---|---|---|
| Barley (*Hordeum vulgare*) | Root rot | 5 | No reports | [36–39] |
| | | 8 | Moderate and high levels of resistance. High resistant transgenic lines | |
| Bean (*Phaseolus vulgaris*) | Root rot | 2 | Moderate and high levels of resistance | [40–43] |
| | | 4 | Moderate and high levels of resistance | |
| | | 5 | Moderate levels of resistance | |
| Cabbage (*Brassica oleracea var. capitata L.*) | Wirestem | 2 | Moderate levels of resistance | [44–46] |
| | | 4 | Moderate levels of resistance | |
| Carrot (*Daucus carota*) | Crown and brace root rot | 1 | Moderate levels of resistance | [47–49] |
| | | 2 | Moderate levels of resistance | |
| | | 4 | Moderate levels of resistance. Moderately resistant transgenic line | |
| Faba bean (*Vicia faba*) | Root rot and stem canker | 2 | Moderate levels of resistance | [50–52] |
| | | 4 | Moderate and high levels of resistance | |
| | | 5 | Moderate levels of resistance | |
| Lettuce (*Lactuca sativa*) | Bottom rot | 1 | No reports | [53] |
| Maize (*Zea mays*) | Banded leaf and sheath blight disease | 1 | Moderate and high levels of resistance | [52,54–57] |
| | | 2 | Moderate levels of resistance | |
| Oat (*Avena sativa*) | Root rot, Bare patch | 8 | Moderate levels of resistance | [38] |
| Oilseed rape (*Brassica napus*) | Root rot and damping-off | 2 | Moderate levels of resistance | [58–60] |
| | | 4 | No reports | |
| Onion (*Allium cepa*) | Stunting | 4 | No reports | [61,62] |
| | | 8 | Moderate levels of resistance | |
| Pea (*Pisum sativum*) | Root rot | 4 | Moderate levels of resistance | [63,64] |
| Potato (*Solanum tuberosum*) | Black scurf and Stem canker | 2 | Moderate levels of resistance | [65–68] |
| | | 3 | Moderate and high levels of resistance | |
| Rice (*Oryza sativa*) | Sheath blight | 1 | Moderate levels of resistance. Moderate levels of resistant transgenic lines. | [69–72] |
| Rye (*Secale cereale*) | Root rot, Bare patch | 8 | Moderate levels of resistance | [38,73] |
| Soybean (*Glycine max*) | Root rot | 1 | Moderate levels of resistance | [74–77] |
| | | 2 | Moderate and high levels of resistance | |
| | | 4 | Moderate and high levels of resistance | |
| Sugar beet (*Beta vulgaris*) | Root rot | 2 | Moderate levels of resistance | [78–80] |
| Tomato (*Solanum lycopersicum*) | Foot and Root rot | 3 | Moderate levels of resistance | [81–83] |
| | Foot and Root rot | 4 | Moderate levels of resistance | |
| Triticale (*Triticale hexaploide*) | Root rot, Bare patch | 8 | Moderate levels of resistance | [38] |
| Wheat (*Triticum aestivum*) | Root rot, Bare patch | 8 | Moderate levels of resistance | [38,39,84,85] |

The genus Fusarium constitutes a sizeable monophyletic group of several hundred species that includes agriculturally important plant pathogens, endophytes, saprophytes, and emerging pathogens of clinical importance [86]. The most important Fusarium species causing root rot is *F. solani* (Table 2). Other *Fusarium* spp. that can cause root rot are

*F. avenaceum, F. graminearum, F. culmorum, F. verticillioides,* and *F. pseudograminearum* (Table 2). However, the latter five are mainly associated with head blight or ear mold in different small-grain cereals [87].

*F. solani* (sexual morph *Nectria haematococca*) is a filamentous fungus of significant agricultural importance. This species is classified as *F. solani* species complex (FSSC) because it contains 60 phylogenetically distinct species [86,88]. Most of the studies on FSSC have been carried out while investigating host-pathogen interactions. Therefore, the group has been subdivided into *formae speciales* (f. sp.) based on host specificity [86]. Phytopathogenic FSSC species include some of the most economically significant plant pathogens associated with root rots and vascular wilts in over 100 crops [89]. Some of the most important *F. solani* that cause problems in agriculture are presented in Table 2.

FSSC causes foot or root rot of the infected host plant, and the degree of necrosis correlates with the severity of the disease [86]. Symptoms on above-ground portions may manifest as wilting, stunting, and chlorosis or lesions on the stem or leaves. However, symptoms vary depending on the specific FSSC pathogen and plant host involved [86].

Other Fusarium species that cause root rots of minor economic importance are *F. chlamydosporum,* which infects coleus and other ornamentals [90,91]. *F. oxysporum* can cause root rot in the Cactaceae family members [92], as well as stem and root rot in melons [93].

**Table 2.** A summary of the main crop species affected by Fusarium root rot.

| Fungi spp. | Crop spp. | Disease Name | Resistance Reported in the Literature | Host Range | Distribution | References |
|---|---|---|---|---|---|---|
| *Fusarium avenaceum* | Alfalfa (*Medicago sativa*) | Root and crown rot | Moderate levels of resistance | Wide range with over 200 species, including pulses, cereals, canola (*Brassica napus*), flax (Linum spp.), and alfalfa (*Medicago truncatula*) | Worldwide | [94] |
| | Barley (*Hordeum vulgare*) | Head blight | Moderate levels of resistance | | | [95,96] |
| | Clover (*Trifolium subterraneum*) | Root rot | Moderate levels of resistance | | | [97] |
| | Pea (*Pisum sativum*) | Root rot | Moderate levels of resistance | | | [98–100] |
| | Wheat (*Triticum aestivum*) | Head blight | Moderate levels of resistance | | | [101–103] |
| *F. culmorum* | Barley (*Hordeum vulgare*) | Head blight | Moderate levels of resistance | Wide range of host plants, including rye (*Secale cereale*), corn (*Zea mays*), sorghum (Sorghum spp.), various grasses, sugar beet (*Beta vulgaris*), bean (Phaseolus spp.), pea (*Pisum sativum*), asparagus (Asparagus spp.), hop (*Humulus lupulus*), strawberry (*Fragaria × ananassa*), and potato (*Solanum tuberosum*) | Worldwide | [104,105] |
| | Oat (*Avena sativa*) | Head blight | Moderate levels of resistance | | | [106] |
| | Wheat (*Triticum aestivum*) | Root rot and head blight | Moderate levels of resistance | | | [101,102,107–109] |
| *F. graminearum* | Barley (*Hordeum vulgare*) | Head blight | Moderate levels of resistance | Wide range, especially many species of cereals and grasses such as oat (*Avena sativa*), rice (Oryza), cucumber (*Cucumis sativus*), soy (*Glycine max*), tomato (Lycopersicon spp.), alfalfa (*Medicago truncatula*), sorghum (Sorghum spp.) | Worldwide | [104,110–112] |
| | Maize (*Zea mays*) | Ear mold and root rot | Moderate levels of resistance | | | [113–117] |
| | Soybean (*Glycine max*) | Pod blight and root rot | High levels of resistance | | | [118–120] |
| | Wheat (*Triticum aestivum*) | Head blight | Moderate levels of resistance | | | [104,121–125] |
| *F. pseudograminearum* | Barley (*Hordeum vulgare*) | Crown rot | Moderate levels of resistance | All major winter cereals barley (*Hordeum vulgare*), oats (*Avena sativa*) and grass genera, such as *Phalaris*, *Agropyron* and *Bromus* | All areas cultivated with wheat and barley | [126,127] |
| | Wheat (*Triticum aestivum*) | Crown rot | Moderate levels of resistance | | | [128,129] |
| *F. solani* f sp. *batatas* | Sweetpotato (*Ipomoea batatas*) | Storage root | Moderate levels of resistance | Not well known | China | [130] |

**Table 2.** *Cont.*

| Fungi spp. | Crop spp. | Disease Name | Resistance Reported in the Literature | Host Range | Distribution | References |
|---|---|---|---|---|---|---|
| *F. solani* f. sp. *glycines* | Soybean (*Glycine max*) | Sudden death syndrome | Moderate levels of resistance | Broad range including bean (Phaseolus spp.), Soybean (*Glycine max*), alfalfa (*Medicago truncatula*), clover (Trifolium spp.), pea (*Pisum sativum*), corn (*Zea mays*), wheat (Triticum spp.), ryegrass (Lolium spp.), pigweed (Amaranthus spp.), and lambsquarters (*Chenopodium album*) | All areas cultivated with soybean in America, Asia, and Africa | [131–133] |
| *F. solani* f. sp. *phaseoli* | Bean (*Phaseolus vulgaris*) | Root rot | Moderate and high levels of resistance | Pea (*Pisum sativum*) | Areas cultivated with bean in all continents except Australia | [134–136] |
| *F. solani* f. sp. *pisi* | Pea (*Pisum sativum*) | Root rot | Moderate levels of resistance | Chickpea (*Cicer arietinum*), clover, soybean, as well as several other non-legume hosts, such as ryegrass (Lolium spp.), potato (*Solanum tuberosum*) ginseng (*Panax ginseng*) and mulberry tree (*Morus alba*) | Worldwide | [137,138] |
| *F. verticillioides* | Maize (Zea mays) | Root and ear rot | Moderate levels of resistance | Wide range of hosts such as rice (*Oryza sativa*), sorghum (Sorghum spp.), soybean (*Glycine max*), alfalfa (*Medicago truncatula*), bean (Phaseolus spp.), wheat (*Triticum spp.*), ryegrass (Lolium spp.) | Worldwide | [139] |

Some species belonging to the genus Phoma are known to cause root rots [140] (Table 3). Although their economic impact is not as significant as the root rots caused by the fungi species mentioned above, considerable yield losses have been reported in alfalfa, sugar beet, corn, and onion (Table 3). *Thielaviopsis basicola* is another global soil-borne fungus that causes black root rot disease. This disease is characterized by necrotic lesions on various parts of the host roots [141–143]. Most reports highlighted the effect of this root rot in cotton (Table 3). Crops such as legumes, tobacco, carrot, citrus, groundnut, and chicory have also been reported to be impacted.

Some other fungal root rot pathogens affect crops with less frequency. These pathogens include *Aspergillus* spp., *Alternaria* spp., *Curvularia* spp., *Rhizopus* spp., and *Penicillium* spp., in fruit trees [144]; *Rigidoporus lignosus* and *Phellinus noxius* in rubber trees [145]; and *Macrophomina phaseolina* in chickpeas [146]. Armillaria root rot is a threat in apples, walnuts, kiwi, and grapes [147,148]. The ascomycete *Rosellinia necatrix* is known to cause white root rot in trees such as apple in the Kashmir valley [149] and avocado in the Mediterranean [150].

**Table 3.** Main crop species affected by fungus Phoma *and Thielaviopsis basicola* root rot.

| Fungi spp. | Crop spp. | Disease Name | Resistance Reported in the Literature | Host Range | Distribution | References |
|---|---|---|---|---|---|---|
| *Phoma betae* | Sugar beet (*Beta vulgaris*) | Crown and root rot | Moderate levels of resistance | Different varieties of *Beta vulgaris* such as table beet, sugar beet, Swiss chard | World-wide distribution, found in all beet-growing areas. | [151] |
| *P. terrestris* (*Setophoma terrestris*) | Corn (*Zea mays*) | Red root rot | Moderate levels of resistance | 45 genera including cereals, vegetables and grasses such as soybean (*Glycine max*), pea (*Pisum sativum*), sugarcane (Saccharum spp.), oats (*Avena sativa*), barley (*Hordeum vulgare*), wheat (Triticum spp.), cucumber (*Cucumis sativus*), tomato (*Lycopersicon* spp.), pepper (*Capsicum annuum*) | World-wide distribution | [152] |
| | Onion (*Allium cepa*) | Pink root rot | Moderate levels of resistance | | | [153,154] |
| *P. sclerotioides* | Alfalfa (*Medicago sativa*) | Brown root rot | Moderate levels of resistance | Wheat (Triticum spp.), barley (*Hordeum vulgare*), and oat (*Avena sativa*). | All areas cultivated with alfalfa in North America and Australia | [155,156] |
| *Thielaviopsis basicola* | Cotton (*Gossypium herbaceum*) | Black root rot | Moderate levels of resistance. Moderate levels of resistance in transgenic lines | Wide range of hosts, plants from at least 15 families including horseradish (*Armoracia rusticana*), carrot (*Daucus carota*), strawberry (*Fragaria* × *ananassa*), tomato (Lycopersicon spp.), bean (Phaseolus spp), pea (*Pisum sativum*) | World-wide distribution | [157–160] |
| | Tobacco (Nicotiana spp.) | Black root rot | Moderate levels of resistance | | | [161,162] |

## 2.2. Oomycetes

Oomycetes resemble fungi in their growth habits and nutritional strategies. However, they are evolutionarily distant from fungi and belong to the kingdom Stramenopiles [163]. Oomycetes are a large group of terrestrial and aquatic eukaryotic organisms. They are dispersed via zoospores, generate thick-walled sexual oospores, possess cellulose in their

cell walls, are vegetatively diploid, have heterokont flagellae (one tinsel and one whiplash), and have tubular mitochondrial cristae [164].

The terrestrial oomycetes are mainly parasites of the vascular plants and include several important pathogens such as *Aphanomyces* spp., *Pythium* spp., and *Phytophthora* spp. that cause root rot (Table 4). These oomycetes appear to have extraordinary genetic flexibility, enabling them to adapt rapidly and overcome chemical control measures and genetic resistance in host plant [165–167].

Among Aphanomyces spp. that cause root rots, *A. cochlioides* and *A. euteiches* cause significant agricultural concerns (Table 4). *A. cochlioides* causes damping off and chronic root rot in sugar beet, spinach, cockscomb, among other species of Chenopodiaceae and Amaranthaceae [168,169] (Table 4). Due to the extended prevalence of the disease in soil and severity in the field, outbreaks of *A. cochlioides* root rot have become a severe problem in many sugar beet growing areas [170]. *A. cochlioides* root rot, when severe, can lead to death and drastically reduced recoverable sugar per ton [171]. Little is known about the genetic basis of resistance to *A. cochlioides* root rot. Still, several sugar beet genotypes have been released and also used in the development of molecular markers that have been found associated with disease resistance genes [170,172,173] (Table 4).

**Table 4.** Main oomycete species that cause root rot disease in crop species.

| Oomycetes spp. | Crop spp. | Reported Symptoms | Resistance Reported in the Literature | Host Range | Distribution | References |
|---|---|---|---|---|---|---|
| *Aphanomyces cochlioides* | Sugar beet (*Beta vulgaris*) | Root rot and damping off. Infected hypocotyl and root rapidly turn black. Undersized plants with yellowed lower leaves. Severely infected plants die. Postharvest reduction in sugar yield | High levels of resistance | Spinach (*Spinacia oleracea*), and several wild species of Beta, including *B. maritima* and *B. patellaris* | Across all sugar beet plantations | [170–174] |
| *Aphanomyces euteiches* | Alfalfa (*Medicago sativa*) | Damping off and root rot. Root tissue becomes honey-brown or blackish-brown. Chlorosis, necrosis, and wilting of the foliage. Severely infected plants die. | High levels of resistance | Faba bean (*Vicia faba*), red clover (*Trifolium pratense*), white clover (*Trifolium repens*), *Medicago truncatula*, lentil (*Lens culinaris*) | All areas cultivated with alfalfa, bean, and pea in Asia, Europe, Oceania, North America | [6,135,175–178] |
| | Bean (*Phaseolus vulgaris*) | | Levels of partial and complete resistance | | | |
| | Pea (*Pisum sativum*) | | High levels of partial resistance | | | |
| *Phytophthora citrophthora* | Citrus spp. | Serious gummosis of citrus trees, root rot, stem necrosis, canker, fruit rot, twig blight, and seedling blight. | Tolerant transgenic *C. sinensis*. Partial levels of resistance in citrus rootstocks. High levels of resistant citrus rootstocks | 88 genera including: kiwifruit (*Actinidia deliciosa*), watermelon (*Citrullus lanatus*) strawberry (*Fragaria ananassa*), walnut (*Juglans regia*), apricot (*Prunus armeniaca*), sweet cherry (*P. avium*), almond (*P. dulcis*), potato (*Solanum tuberosum*), cocoa (*Theobroma cacao*), blueberries (*Vaccinium*) | Worldwide | [179–184] |
| *Phytophthora nicotianae* | Citrus spp. | Symptoms vary per host. Damping-off, crown rot, leaf blight, fruit rot. Occasionally, it attacks aerial parts of the plant and can cause brown rot of fruit. | Tolerant transgenic *C. limonia*. Partial levels of resistance in citrus rootstocks | 255 genera in 90 families. including tobacco, citrus, cotton, and orchids | Worldwide | [180,181,185–189] |
| | Tomato (*Solanum lycopersicum*) | | Partial levels of resistance | | | |

**Table 4.** *Cont.*

| Oomycetes spp. | Crop spp. | Reported Symptoms | Resistance Reported in the Literature | Host Range | Distribution | References |
|---|---|---|---|---|---|---|
| *Phytophthora cactorum* | Apple (*Malus domestica*) | Damping off of seedlings, fruit rot, leaf, stem and root rot, collar and crown rot, stem canker. | Partial levels of resistance | 154 genera of vascular plants in 54 families | Worldwide | [181,190–195] |
| | Strawberry (*Fragaria* × *ananassa*) | | Partial levels of resistance | | | |
| *Phytophthora cinnamomi* | Avocado (*Persea americana*) | Root rot, heart rot, wilt. Primary infection at the feeder roots, resulting in a brownish black and brittle appearance. | Partial levels of resistance | 266 genera in 90 families, commonly hardwood trees. | Worldwide | [181,196] |
| *Phytophthora fragariae* | Rasberry (Rubus spp.) | Red stele or red core root rot. Symptoms also include wilting of leaves, reduced flowering, stunting, and bitter fruit | Partial and high levels of resistance | - | All areas cultivated with rasberry and strawberry in Asia, Australia, New Zealand, Europe, North America | [181,197–200] |
| | Strawberry (*Fragaria* × *ananassa*) | | Partial levels of resistance | | | |
| *Phytophthora sojae* | Soybean (*Glycine max*) | Root and stem rot; pre- and post-emergence damping-off, seedling wilt, seedling blight. Plant may turn reddish-orange to orange-brown in color. | Partial levels of resistance | Lupine (Lupinus spp.); also reported in six other genera in five families | All areas cultivated with soybean in Australia, North America (Canada, USA)., South America (Chile), Asia (Korea, China) and New Zealand | [181,201,202] |
| *Phytophthora capsici* | Pepper (*Capsicum annuum*) | Fruit, stem, and root rot., seedling damping-off, and leaf wilt. Leaf tissue becomes wilted, light green or gray-green, and later tan to white. Fruit rots are olive green or light green in color. | Partial levels of resistance | 51 genera in 28 families, including tomatoes (*Lycopersicon esculentum*), other Solanaceae spp., Macadamia spp., cacao (*Theobroma cacao*) | Worldwide | [181,203–205] |

**Table 4.** *Cont.*

| Oomycetes spp. | Crop spp. | Reported Symptoms | Resistance Reported in the Literature | Host Range | Distribution | References |
|---|---|---|---|---|---|---|
| *Phytophthora medicaginis* | Alfalfa (*Medicago sativa*) | Root rot, damping-off of seedlings. Reddish-brown or black root lesions. Mature plants exhibit chlorosis, desiccation of foliage, and reduced growth but may also collapse. | High levels of resistance | Sainfoin (*Onobrychis viciifolia*), chickpea (*Cicer arietinum*), cherry (*Prunus mahaleb*) | Cosmopolitan, throughout the range of the host | [181,206–210] |
| | Chickpea (*Cicer arietinum*) | | Partial levels of resistance | | | |
| | Soybean (*Glycine max*) | | Partial levels of resistance | | | |
| *Pythium ultimum* | Bean (*Phaseolus vulgaris*) | Disease can manifest as seed rot, preemergence and postemergence damping-off, root rot, dark brown or reddish roots, and sunken lesions on lower hypocotyls. Plants are stunted or chlorotic. Root tips of diseased plants appear as brown. | High levels of resistance | Cabbage (*Brassica oleracea* var. *capitata*), carrot (*Daucus carota*), melon (Cucumis melon), wheat (*Triticum aestivum*) | Worldwide | [211–215] |
| | Cucumber (*Cucumis sativus*) | | No records | | | |
| | Sorghum (*Sorghum bicolor*) | | No records | | | |
| | Soybean (*Glycine max*) | | Partial levels of resistance | | | |
| | Sugar beet (*Beta vulgaris*) | | Partial levels of resistance | | | |
| | Tomato (*Solanum lycopersicum*) | | No records | | | |

<div align="center">

**Table 4.** *Cont.*

</div>

| Oomycetes spp. | Crop spp. | Reported Symptoms | Resistance Reported in the Literature | Host Range | Distribution | References |
|---|---|---|---|---|---|---|
| *Pythium irregulare* | Clover (*Trifolium subterraneum*) | Pre- and post-emergence damping off of seedlings (greenhouse) and root rot (field) of older plants. Contaminated seeds and seedlings will quickly turn brown and soft before decomposing. Foliage may turn chlorotic or a greenish-grey and wilt. | Moderate resistance | Over 200 host species including pineapple (*Ananas comosus*), cereals, grasses, celery (*Apium graveolens*), pepper (*Capsicum annuum*), pecan trees (*Carya illinoinensis*), Citrus spp, strawberries (*Fragaria × ananassa*), lentils (*Lens culinaris*), corn (*Zea mays*), soybean (*Glycine max*), cucumber (*Cucumis sativus*), onion (Allium cepa), carrot (*Daucus carota*) and a number of floricultural crops | Cosmopolitan in greenhouses and field systems | [97,216–218] |
| | Soybean (*Glycine max*) | | Moderate to high levels of resistance | | | |
| *Pythium aphanidermatum* | Cucumber (*Cucumis sativus*) | Causes root and stem rots, as well as pre- and post- emergence damping off. It causes blights of grasses and fruit. Roots are blackened, mushy and rotten. It causes wilting, loss of vigor, stunting, chlorosis and leaf drop. Beets and other fleshy plant organs are susceptible to rot in the field and during storage. | No reports | Broad host range, including cotton (Gossypium spp.), grasses, papaya (*Carica papaya*), cereals, Brassica species and beans (*Phaseolus vulgaris*) | Cosmopolitan in greenhouses and field systems | [219–223] |
| | Lettuce (*Lactuca sativa*) | | No reports | | | |
| | Pepper (*Capsicum annuum*) | | Moderate levels of tolerance | | | |
| | Soybean (*Glycine max*) | | High resistance | | | |
| | Sugar beet (*Beta vulgaris*) | | Partial levels of resistance | | | |
| | Tomato (*Solanum lycopersicum*) | | No reports | | | |

*A. euteiches* causes seedling damping-off and root rot disease in a variety of field crops worldwide. It first causes softened and water-soaked roots that result in stunted seedlings and yellow leaves. Then the pathogen spreads rapidly, and the cortical tissue and the delicate branches of feeding rootlets are destroyed. In severe cases, the plants collapse and die (Table 4) [3,20].

Main yield losses caused by *A. euteiches* are observed in legumes. *A. euteiches* is the most devastating pathogen of pea in several countries, with yearly losses that average 10 to 80% each year [3,6]. Significant yield losses have also been reported in alfalfa [224], clover [225], fava beans [226], and lentils [227]. Like *A. cochlioides*, *A. euteiches* is a strictly soil-borne pathogen that may survive up to 10 years in soil [29], and no efficient chemical control is currently available. The only way to control the disease is to avoid cultivating legumes in infested fields for 10 years [3,29]. To date, no fully resistant pea cultivars have been developed. In 2012, eight pea germplasm lines, obtained via selective breeding, carrying partial resistance to *A. euteiches* and acceptable agronomic characteristics were released for fresh, frozen, and dry pea production (Table 4) [177]. Resistant lines of alfalfa are also available to growers [175,176].

Pythium genus possesses over 200 described species, and at least 10 Pythium spp. cause Pythium damping-off and root rot in various legumes and monocots (Table 4). Pythium root rot infection symptoms are similar to other root rots; however, only the root tips show necrosis during early infection [228]. It is also typical of this pathogen that the entire primary root's rapid black rot moves up to the stem [34]. *P. ultimum* and *P. irregulare* have been reported as the most ubiquitous pathogens in this group, regularly found in the field, sand, pond and stream water, and decomposing vegetation [34,228]. In the greenhouse industry, the three most commonly encountered root rot Pythium species are *P. ultimum*, *P. irregulare*, and *P. aphanidermatum* [228].

*P. ultimum* is a principal causal agent of seed decay and pre- and post-emergence damping-off in beans [215,229]. A study found that only cream-seeded beans exhibited high resistance levels, while all the white-seeded accessions were susceptible [215]. *P. irregulare* is often the most common pathogenic species of Pythium in soybean farms [230,231]. These studies found that *P. irregulare*, compared to other Pythium *spp.*, had the highest pathogenicity levels in soybean. A total of 65 soybean genotypes were evaluated for resistance to *P. irregulare*, and about a third showed moderate to high levels of resistance (Table 4) [216].

*Pythium aphanidermatum* is the predominant pathogen in greenhouse-grown cucumber. It can rapidly spread through zoospores in a recirculating nutrient film culture system [232–234]. *P. aphanidermatum* is also one of the most critical sugar beet diseases in temperate areas with high soil moisture levels. In addition to direct damage to plants in the fields, this pathogen also causes root rot in storage [235]. Several sugar beet genotypes have been found to be partially resistant to *P. aphanidermatum* root rot (Table 4) [220,235]. Other economically important plant species affected by Pythium spp. are parsnip and parsley [236], wheat [237], and sugarcane [238]. *P. aphanidermatum* and *P. ultimum* mediated root rot has been reported in ornamental plants [239].

*Phytophthora* spp. represent more than 100 species, and most of them have been classified as aggressive plant pathogens that cause extensive losses in agricultural and horticultural crops [240]. Phytophthora means "plant destroyer," a name coined in the 19th century when the potato disease caused by *Phytophthora infestans* (causal agent of potato late blight) set the stage for the Great Irish Famine [241]. Phytophthora causes extensive tuber damage and also impacts above-ground parts of the plant in potato. General symptoms of *Phytophthora* infection include wilting, yellow or sparse foliage, and branch dieback [4].

*P. citrophthora* is the most wide-spread oomycete pathogen in citrus growing areas accounting for millions of dollars in crop losses annually [242,243]. In citrus, *P. citrophthora* causes gummosis, root rot, and during winter, it causes brown rot of the fruit. *P. nicotianae* also causes foot rot and root rot in citrus. This pathogen is more commonly found in

subtropical areas of the world [243,244]. Nursery- and large-trees can be rapidly girdled and killed by both pathogens [243].

*P. cactorum* is another pathogen capable of producing high yield losses in fruit trees. In Canada and certain US regions, it has been identified as the most important cause of crown rot of apple [190]. The use of rootstocks resistant to *P. cactorum* and other Phytophthora spp. has been considered a good management practice. Sources with different resistance levels have been identified since 1959; however, no highly resistant rootstocks have been found (Table 4) [190,192,194,195]. In strawberry, *P. cactorum* crown rot is also considered a disease of commercial importance worldwide. Several strawberry lines with partial resistance to the disease have been identified recently (Table 4) [191,193].

*P. cinnamomic* is another Phytophthora pathogen that affects fruit trees. This disease causes problems mainly in avocado. On average, this disease leads to an annual loss of 10% of the world's avocado crop [245]. This disease has eliminated commercial avocado production in many Latin American regions and is the major limiting factor of production in Australia, South Africa, and California [196]. Therefore, the development of resistant *P. cinnamomic* rootstocks is currently one of the most important goals for the avocado industry. *P. cinnamomi* is also a problem for pineapple production in Australia since it reduces plant growth and yield. *P. cinnamomi* root rot may result in total loss of this crop, especially for the new pineapple hybrids, which are susceptible to *P. cinnamomic* [246]. Other Phytophthora species that significantly affect agriculturally important crops are *P. fragariae*, *P. sojae*, *P. capsici*, and *P. medicaginis* (Table 4).

### 2.3. Bacteria and Viruses

Bacteria are not a significant root rot causing agent. However, these root rots can cause substantial economic damage. Main yield losses occur in potato and sweet potato. Considerable losses have also been reported for green peppers and Chinese cabbages (*Brassica campestris* subsp. *pekinensis*) [4,7–10,247].

Bacteria commonly gain entry into the host through wounds in the roots [4,248]. They may also be able to gain access through the leaves, where bacteria develop under aerobic conditions in the aerial parts or migrate to the bulb, rhizome, or directly infect the storage organ [4,247]. These bacteria are characterized by the production of large quantities of extracellular enzymes that include pectinases, cellulases, proteases, and xylanases, which digest the host cell walls and cause disease [249,250]. From this set of enzymes, the pectinases are believed to be the most important in pathogenesis, causing tissue maceration and cell death [247]. The ability to produce a broader range of enzymes more rapidly and larger quantities than pectolytic saprophytic microorganisms enables bacterial root rots to invade living plants more readily [247,251].

The number of identified bacterial root rot pathogens belong to two genera, Pectobacterium and Dickeya. Overall, four pathogens, *Pectobacterium carotovorum* subsp. *Carotovorum* (formerly *Erwinia carotovora* subsp. *Carotovora*), *P. atrosepticum* (formerly *E. carotovora* subsp. *Atroseptica*), *Dickeya dianthicola*, and *D. solani* (both previously known as *E. chrysanthemi*), cause wilt and rot diseases in monocot and dicot plants worldwide. Of these pathogens, *P. carotovorum* subsp. *Carotovorum* has the broadest host range worldwide. *P. atrosepticum* is restricted to potato. *D. dianthicola* and *D. solani* are pathogenic to many plants in the tropical and subtropical regions, and affect maize and dahlia in the temperate regions.

Symptoms of bacterial root rot, mainly characterized in potato and sweet potato, include chlorosis of leaf tissue and a black, water-soaked decay at the bottom of the stems that gradually extends to the top. In severe cases, the entire plant collapses [252]. Fibrous roots have localized lesions with a characteristic black appearance. In storage roots, sunken brown lesions with black margins can be observed at the surface [253].

The information regarding viruses as the causal agents of root rot is limited. Some studies have reported the effect of Cassava brown streak virus (CBSV) [11,12] and Ugandan cassava brown streak virus (UCBSV) [254] in root rot development. Several scions of elite

breeding lines have been identified as resistant to both viruses [255,256]. However, sparse or no evidence is available in yield losses and molecular mechanisms of resistance within the host. Transgenic cassava lines expressing interfering RNAs against the sequence of the CBSV and UCBSV have increased the level of resistance against these two viruses [257–259]. These transgenic lines provided proof of principle for the control of CBSV and UCBSV. Information regarding these non-traditional root rot agents is expected to increase as detection methods evolve [4].

## 3. Molecular Mechanisms of Resistance Against Root Rot Pathogens

When a pathogen attacks a plant, several molecular mechanisms are activated. Plants first respond to pathogen infection via pathogen-associated molecular pattern (PAMP) triggered immunity (PTI) [260]. Pathogens respond with a variety of effector proteins to counter the PTI [261]. Plants can sometimes detect these effectors and respond with the commonly known effector-triggered immunity (ETI).

Most plant–pathogen interactions characterized so far fall under the gene-for-gene interaction model. According to this model, a dominant or semi-dominant resistant (R) gene from the host and a corresponding avirulence (AVR) gene from the pathogen interact and activate further downstream reactions (PTI or ETI). The R–AVR interaction concludes with an incompatible response in which no disease symptoms are produced [262]. The most abundant *R* genes code for the plant nucleotide binding site leucine-rich repeat (NBS-LRR) proteins that are responsible for detecting potential pathogens and triggering a defense response [263].

Lack of corresponding R–ARV interactions leads to a compatible response leading to pathogenesis. Both compatible and incompatible interactions result in the recruitment of different sets of proteins that determine total and partial resistance or susceptibility.

The molecular mechanisms of root rot pathogens-mediated interactions are very diverse. The variations in plant–pathogen interaction are dependent on the species and race of the pathogen and the specific host genotype involved. The following sections describe validated and proposed molecular defense mechanisms against fungal—Rhizoctonia, Fusarium, *Phoma*, and *Thielaviopsis basicola*; and *oomycete—Aphanomyces*, Pythium, and Phytophthora, root rot pathogens.

## 4. Common Causal Agents of Root Rots

### 4.1. Fungal Root Rot

#### 4.1.1. Rhizoctonia Solani

*Rhizoctonia solani* is a pathogen with a broad host range. Resistance to *R. solani* has been studied mainly in sugar beet and rice with additional reports in potato and model organisms, such as Arabidopsis and tobacco.

In sugar beet, the genetic basis for Rhizoctonia resistance is considerably narrow. The GWS 359-52R genotype is the universal parental line for essentially all resistant cultivars [264–266]. An early study suggested that the resistance to *R. solani* is associated with at least two loci with two or three alleles [267]. This hypothesis was supported by a recent quantitative trait locus (QTL) analysis, which localized the resistance loci on chromosomes 4, 5, and 7. These QTLs collectively explain 71% of the total phenotypic variation [266]. Genes involved in pathogen recognition and responses downstream of R-genes co-segregated with the resistance QTL located on chromosomes 4 and 7, respectively [266]. Genes that show similarity with the Xa21 and Pto were found to co-segregate with the QTL on chromosome 5. Xa21 is a well-characterized cell membrane receptor that, through phosphorylation and cleavage of its intracellular kinase domain, perceives the presence of pathogens [268]. Xa21 relays the signal to the nucleus through multi-step signal cascades, involving mitogen-activated protein kinase (MAPK) and WRKY signaling [269,270]. Xa21 is known to trigger hormone signaling, especially cytokinins [268]. The role of cytokinins in defense response remains elusive, however, studies have reported that cytokinins prompt salicylic acid (SA) accumulation [271,272]. The Pto gene encodes

a cytoplasmic serine-threonine kinase that interacts with avirulence proteins and confers HR-mediated resistance [273]. Pto has been considered an important candidate gene for broad-spectrum resistance in molecular breeding approaches [274,275].

Metabolomic profiling aided in characterizing the changes in sugar beet 0 and 7 days after inoculation (dai) with *R. solani* [276]. N1-caffeoyl-N10-feruloylspermidine and codonocarpine, both alkaloids, showed higher levels in resistant germplasm roots than susceptible germplasm at 0 dai. The role of alkaloids has been suggested to be a conserved defense response to *R. solani* and other necrotrophic fungal pathogens [276,277]. N1-caffeoyl-N10-feruloylspermidine and codonocarpine alkaloids have multiple and complex roles, therefore, it is difficult to predict their specific function against *R. solani*. Furthermore, two oleanic acid-like compounds (saponins) were found in the resistant germplasm, and their abundance continued to increase after infection with *R. solani.* Saponins are known to have antifungal activity [278,279]. Thus, three metabolites, N1-caffeoyl-N10-feruloylspermidine, codonocarpine, and oleanic acid-like compounds, are important candidates for follow-up studies on the interaction between sugar beet and *R. solani*.

No rice cultivar shows complete resistance, but partial resistance to *R. solani* has been reported. These studies have proposed that different defense mechanisms are activated in the partially resistant rice genotypes. A summary of changes detectable following *R. solani* inoculations in the partially resistant rice genotypes is presented in Figure 1.

In total, 25 genes were found to be differentially expressed in rice after infection with *R. solani* [280]. These same genes were also differentially expressed when rice was challenged with *Magnaporthe grisea* and *Xanthomonas oryzae,* suggesting a conserved defense response to different pathogens. This analysis showed that Pathogenesis-Related (PR) 1b and probenazole-inducible protein 1 (PBZ1) genes were detected at 12 h post-infection (hpi) when the *R. solani* mycelium started to grow on the surface of the plant [280]. The expression of PR1b increased gradually from 12 to 72 hpi. A few lesions began to develop at 36 hpi, and typical lesions developed at 48 hpi. Meanwhile, the expression of PBZ1 increased to its maximum level at 48 hpi. PR1b gene is induced by pathogens commonly associated with SA-related systemic acquired resistance (Figure 1) [281,282]. Further downstream function or signaling effects of the PR1b protein remain unknown. The PBZ1 gene, a PR10 family protein, has been shown to induce cell death in rice, *Nicotiana tabacum*, and Arabidopsis lines [283]. Cell death is caused by PBZ1-RNase activity inside the plant cell (Figure 1) [283,284]. On the other hand, the gene glutathione peroxidase 1 (GP1), which protects cells against both oxidative stresses and inhibits oxidative stress-induced cell death, was found to be induced at 4 hpi, reaching a maximum at 24 hpi upon *R. solani* infection (Figure 1) [285]. Feedback signaling potentially provides an equilibrium between the antagonistic action of PR1b and PBZ1 versus GP1 during the defense response against *R. solani.*

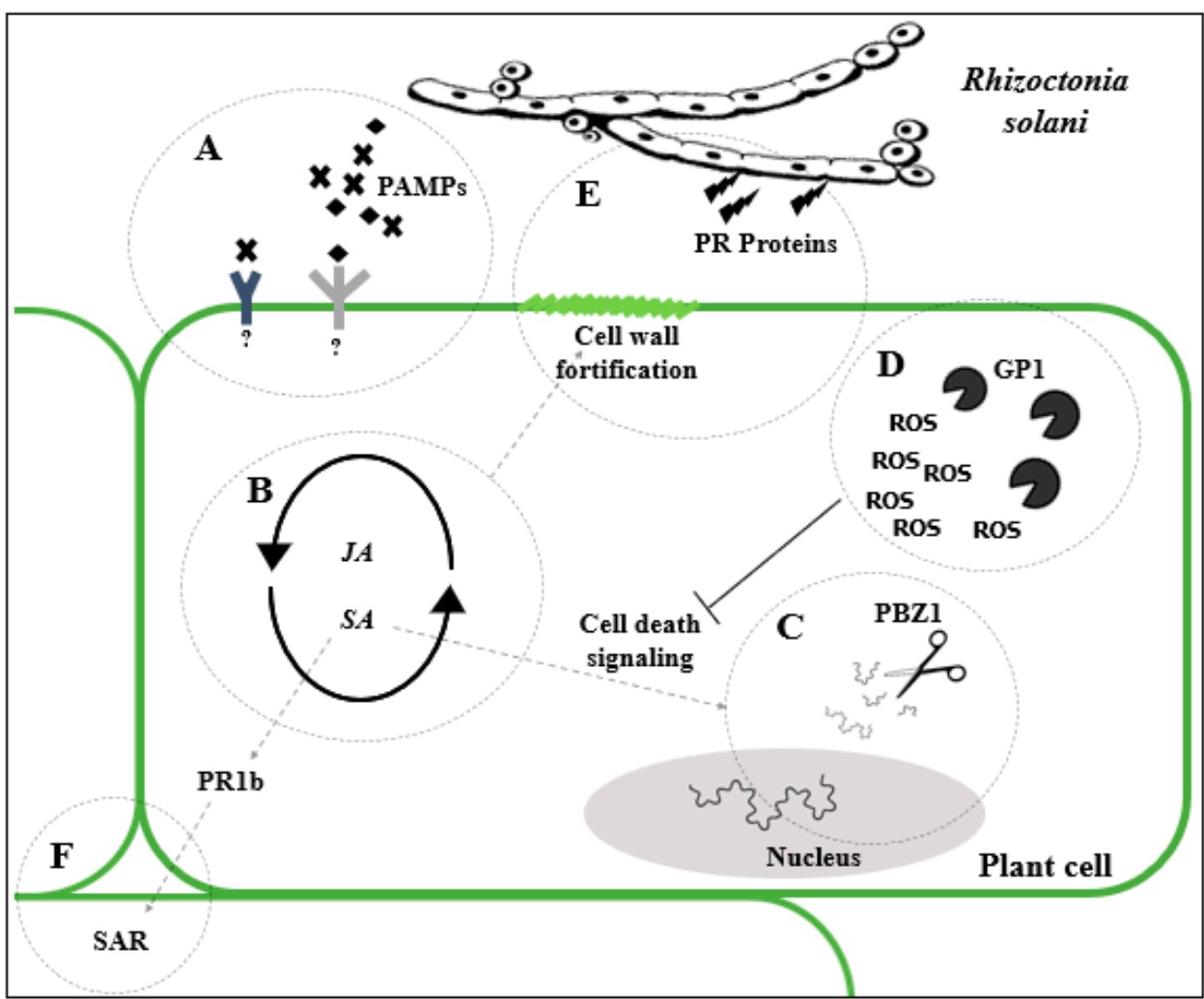

**Figure 1.** A model representing a resistance response following inoculation with *Rhizoctonia solani* in rice. Events include the recognition of pathogen-associated molecular patterns (PAMPs) by the host (**A**); cell signaling induced by both jasmonate (JA) and salicylic acid (SA) hormones (**B**); cell death (**C**); Reactive oxygen species (ROS) scavenging (**D**); synthesis and action of enzymes that attack the pathogen, as well as prepare the host for the attack (**E**); and systemic acquired resistance (**F**). The specific cell receptors participating in recognition of PAMPs are still unknown. SA participates directly in cell death and systemic acquired resistance (SAR) activation, which are triggered by the PBZ1-RNAse and PR1b enzymes, respectively. ROS scavenging is performed by the glutathione peroxidase 1 (GP1), which protects cells against both oxidative stresses and inhibits oxidative stress-induced cell death. JA triggers the activation of the phenylpropanoid pathway for the lignification of the host cell walls. Chitinases are synthesized and released to combat *R. solani*. Defense against *R. solani* is best achieved by early action against the young hyphae. Model derived from the results presented in S. Chen et al., (2004); Shrestha et al., (2008); Taheri and Tarighi, (2010); C.-J. Zhao et al., (2008) [280,285–287].

Another study demonstrated that chitinase levels correlated with resistance to *R. solani* in rice cultivars (Figure 1) [286]. Chitinase activity was detected 24 h after inoculation of seven moderately resistant cultivars. However, in a susceptible genotype, chitinase activity was delayed and was seen only after 36 h post-inoculation. Moderately resistant rice cultivars had higher levels of chitinase activity and lower disease severity and number of infection cushions formed than the susceptible genotype [286]. Resistance to *R. solani* in rice has also been associated with the jasmonate (JA) mediated priming of the phenylpropanoid pathway and the resultant enhanced lignification (Figure 1) [287]. A gene that rapidly accumulated to high, sustained levels in rice after *R. solani* challenge was the disease resistance response protein 206 [280]. This gene participates in the production of active lignans, thus playing a central role in plant secondary metabolism. It was proposed

that it be worth evaluating this protein's role in defense against *R. solani* in rice [280]. Studies indicate that protection in potatoes against *R. solani* is enhanced by co-expression of chitinases, 1,3-β-glucanases, and osmotin proteins [288,289]. It has been hypothesized previously that co-expression of these enzymes is needed to speed up the destruction of *R. solani* hyphae [289,290]. Newly synthesized chitin in cell walls of young hyphae is more sensitive to enzymatic degradation [291]. Therefore, the defense against *R. solani* is best achieved by early action against the young hyphae.

Resistance response in potato, bean, and cowpea seems to be dependent on SA [289,292,293]. On the other hand, a screening of 36 *Arabidopsis thaliana* ecotypes with differences in auxin, camalexin, SA, abscisic acid (ABA), and Jasmonic acid (JA_–ethylene pathways did not reveal any variation in response to *R. solani*. It demonstrated that resistance to *R. solani* was independent of these metabolic pathways [294]. In *A. thaliana*, it has been shown that NADPH oxidases mediate the resistance to *Rhizoctonia solani* [294]. The NADPH oxidase double mutant resulted in an almost complete loss of resistance. This last observation highlights a unique target to be evaluated or incorporated in crop plants such as sugar beet and rice.

### 4.1.2. Transgenic Approach to Combat *Rhizoctonia solani*

A polygalacturonase-inhibiting protein (PGIP) from sugar beet introduced into *Nicotiana benthamiana* resulted in enhanced resistance to *R. solani* [295]. Crude PGIP protein extracts from transgenic *N. benthamiana* plants significantly inhibited *R. solani* polygalacturonase. The crude extracts also inhibited polygalacturonase from *Fusarium solani* and *Botrytis cinerea*. Transgenic plants were also significantly more resistant to these three fungi [295]. Similarly, the expression of a common bean-PGIP also conferred strong resistance against *R. solani* in tobacco [296]. Transgenic tobacco expressing the bean PGIP also expressed enhanced resistance against *Phytophthora parasitica* and *Peronospora hyoscyami* [296]. Transgenic sugar beet expressing the bean PGIP gene showed only minor quantitative effects in enhancing resistance against *R. solani* [297].

Transgenic rice expressing 1-aminocyclopropane-1-carboxylic acid synthase (ACS2, a key enzyme of Ethylene biosynthesis) gene exhibited increased resistance to a field isolate of *R. solani*, as well as different races of *M. oryzae* [298]. This study showed an increased expression of PR1b (10 to 60-fold) and PR5 (2.0 to 7.9-fold) genes in the transgenic lines, as well as no negative impact on crop productivity [298]. Rice transgenic lines expressing broad-spectrum resistance 2 (BSR2) [299], thaumatin-like proteins [69], and a chitinase [300] have also exhibited enhanced resistance against *R. solani* as well.

### 4.1.3. *Fusarium solani* Root Rot: The Case of Pea and Similitudes with Soybean

One of the predominant causal agent of root rots in *P. sativum* is *Fusarium solani* f. sp. *pisi* (*Fsp*). The molecular responses to *Fsp* infection have been reported in pea since the late 1970s. A model of partially resistant and susceptible reactions against *Fsp* in pea is presented in Figure 2. Some of these studies have reported the association between pea and its non-host pathogen *F. solani* f. sp. *phaseoli* (*Fsph*). Generally, non-host resistance is more durable due to the involvement of multiple mechanisms making it an important model to study [301].

Experiments examining the interaction between pea-*Fsph* and pea-*Fsp* showed that *Fsph* and *Fsp* elicitors such as chitosan and DNase are released and directly affect the chromatin structure of the plant host (Figure 2) [302–305]. In turn, chromatin structure changes result in the alteration of gene expression patterns (Figure 2). Changes in chromatin structure, such as the decrease in the expression of High-Mobility Group (HMG) A transcription factor and modification of histones H2A and H2B, have been temporarily associated with the onset of PR gene activation (Isaac et al. 2009) (Figure 2). Pretreatment of pea tissue with chitosan and *Fsph* DNase has been shown to enhance protection against *Fsp* [306,307].

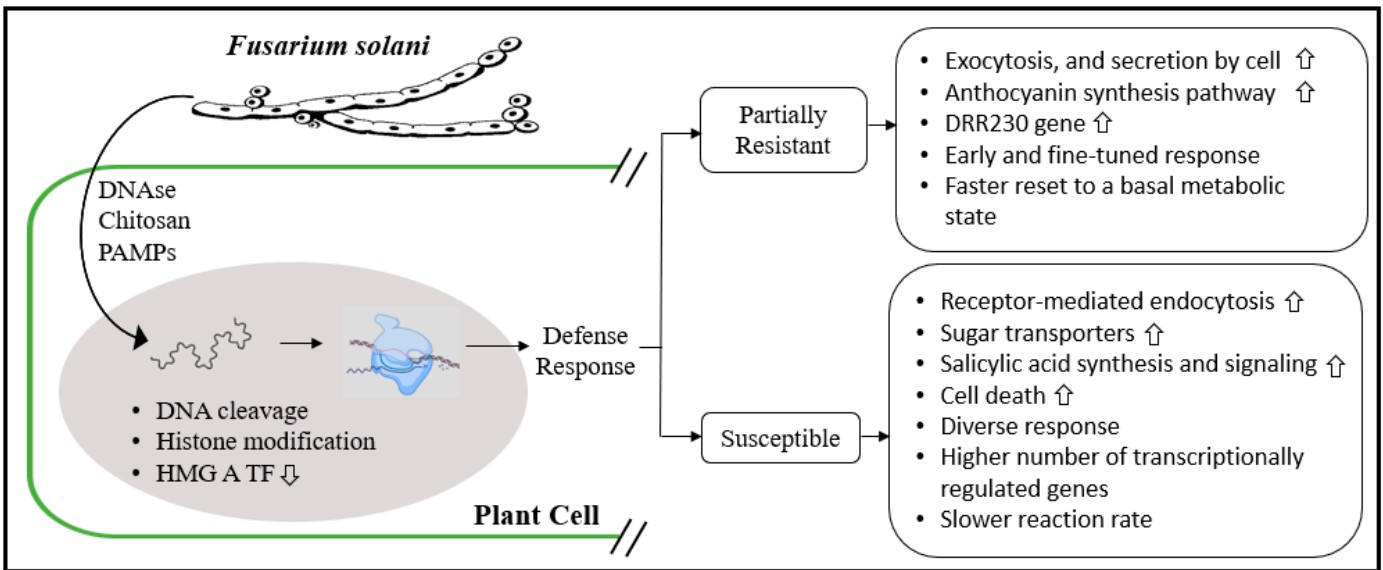

**Figure 2.** A model representing reported changes detected following *Fusarium solani* f. sp. *pisi* (*Fsp*) inoculation in pea. Events include the action of *Fsp* DNAse, chitosan, and PAMPs and/or their detection by the host. DNase and chitosan are associated with nuclear fragmentation in the plant nucleus affecting chromatin structure. These changes, along with the host's detection of PAMPs, trigger defense responses such as the accumulation of pathogenesis-related (PR) genes. Specific responses in partially resistant and susceptible pea genotypes are depicted. Upwards and downwards pointing arrows represent overexpression and underexpression of genes, respectively. Model developed from the results presented in Hadwiger, (2008); B. A. Williamson-Benavides et al., (2020) [304,308].

The accumulation of PR RNA seems crucial to acquiring resistance against *Fsph* [304]. PR proteins, such as the defensins disease-resistance response 230 (DRR230) and DRR39, and the RNAse PR-10 have a direct antifungal effect [305,309]. Other PR proteins, such as PR-1, a homolog of PR1b in Arabidopsis, act as positive regulators of plant immunity [305]. Chitinase and β-glucanase are constitutively expressed, but their basal expression increases 10 h post-inoculation with *Fsph* [310]. These PR proteins' expression occurs within the crucial period for developing a resistance response against *Fsph* [303]. Similar mechanisms of resistance are triggered in pea to halt *F. oxysporum* pv. *pisi* infection [304,311].

There are significant similarities in the biochemical responses induced by the non-host pathogen *Fsph* and the host-pathogen *Fsp* in pea. In both cases, there is a nearly complete suppression of the phosphorylation of chromatin proteins, which leads to the elimination of HMG A from the cell nuclei and alteration of the histone biochemical structure [302–305]. Additionally, the same PR genes, such as DRR230, DRR39, RNase PR-10, and PR-1, seem to be upregulated when challenged with the two pathogens. The major difference in the biochemical responses induced by *Fsph* and *Fsp* is the speed at which the plants react. The type of response exhibited by pea varies with the rate of induction of PR genes and other associated biochemical pathways. In case of either the *Fsph* or *Fsp* infection, the fungus releases DNAses extracellularly, which localize to the host nuclei and degrades the nuclear DNA (Figure 2) [304,305,312]. Fungal DNases can also impact the nuclei in the fungal mycelia and trigger their deterioration [304]. In case of a compatible interaction between *Fsp* and a pea genotype, the host's slower reaction rate allows *Fsp* to protect a small number of its nuclei from fungal DNAses, allowing the growth of *Fsp* to resume after 12 h post-inoculation [305,307]. In contrast, the relatively rapid response generated in the host against *Fsph* terminates the fungi's development at 6 h post-inoculation [304,305].

In pea, phenylalanine ammonia-lyase and chalcone synthase enzymes are upregulated two hours post-inoculation with *Fsph* and *Fsp*. These two enzymes participate in the phenylpropanoid pathway and play a significant role in producing flavonoids and isoflavonoids (Figure 2). The phenylpropanoid pathway potentially plays a role in partial

resistance to *Fsp* in pea (Figure 2) [308] and partial resistance to *Fusarium solani* f. sp. *glycines* (*Fsg*) in soybean [313,314].

Pea contains an isoflavone synthase enzyme, which redirects phenylpropanoid pathway intermediate naringenin (4′,5,7-trihydroxyflavanone) to synthesize pisatin (Sreevidya et al. 2006). Pisatin is an extensively studied phytoalexin from pea, and its production increases in the presence of *Fsp*, *Fsph*, and chitosan [302]. Interestingly, *Fsp* isolates incapable of demethylating pisatin are low in virulence and susceptible to pisatin [304,315]. The phytoalexin glyceollin levels increased in the *Fsp*-inoculated roots of two partially resistant soybean cultivars compared to a susceptible one [313]. The role of these two phytoalexins in the interaction between plant host and *Fusarium* remains to be elucidated.

A time-course RNAseq compared the expression of *Fsp*-responsive genes in four partially resistant and four susceptible pea genotypes after 0, 6, and 12 h post pathogen challenge [308]. *Fsp* challenge produced a more intense and diverse overexpression of genes in the susceptible genotypes. In contrast, the partially resistant genotypes showed fewer changes in the expression of defense-related genes and a faster reset to a basal metabolic state (Figure 2). In the partially resistant genotypes, gene expression and Gene Ontology (GO) enrichment analyses revealed that genes involved in exocytosis and secretion by cell, the anthocyanin synthesis pathway, as well as the DRR230 PR gene were overexpressed (Figure 2) [308]. Genes coding for receptor-mediated endocytosis, sugar transporters, SA synthesis, and signaling, and cell death were overexpressed in the susceptible genotypes (Figure 2).

A total of five recombinant inbred line (RIL) populations have been analyzed to identify QTLs in response to *Fsp* challenge [316–319]. A major QTL, named *Fsp-Ps2.1*, has been found on chromosome 6 that explains 39.0 to 53.4% of the phenotypic variance [316–318]. The A (pigmented flower and anthocyanin pigmentation) gene was mapped within the interval of *Fsp-Ps2.1*. However, *Fsp-Ps2.1* was mapped in a white flower (*aa* x *aa*) cross [317]. Therefore, it has been hypothesized that the resistance gene(s) responsible for *Fsp-Ps2.1* effect may not necessarily be *A* since *Fsp-Ps2.1* was initially identified in a white (*a*) flowered cross. *Fsp-Ps2.1* co-located with the Aphanomyces root rot partial resistance QTL *Ae-Ps2.1* [316,317]. A second QTL, *Fsp-Ps6.1*, explained 17.3% of the phenotypic variance. In total, three defensin family genes, pI39 and DRR230-A and DRR230-B, were mapped near *Fsp-Ps3.1* [317]. A different subset of parental white-flowered genotypes was crossed to developed two populations that segregate for *Fsp* resistance [319]. QTL analysis of these two populations identified five QTLs that explain 5.26 to 14.76% of the resistance to *Fsp*. Overall, three of these are considered newly reported QTLs. The recently identified QTLs and the absence of a major QTL on chromosome 6, reported in previous studies, reflects the wide degree of genetic resources of resistance available to combat *Fsp* in pea [319].

### 4.1.4. Transgenic Approach to Counter *Fusarium solani* Root Rot

An antibacterial peptide-encoding gene from alfalfa seeds, alfAFP, was fused to the C-terminal of the rice chitinase-encoding gene and introduced into tobacco [320]. The recombinant protein enhanced resistance against *F. solani* in transgenic tobacco plants. Transgenic lines did not exhibit wilting symptoms, even 30 days post-inoculation with *F. solani* [320]. In a different approach, the Ethylene-responsive factor *ERF94* was expressed in potatoes. The transgenic lines exhibited enhanced resistance to *F. solani* [321]. Transgenic potato plants showed a limited production of $H_2O_2$ and increased expression of antioxidant enzymes and PR proteins [321].

### 4.1.5. *Fusarium graminareum* Root Rot

As is the case with other root rots, the mechanism of interaction of soybean and *Fusarium graminareum* has not been studied in depth. However, several QTL analyses have identified the potential loci responsible for resistance, and in some cases, several candidate genes have been mapped to these QTL regions.

A total of five putative QTLs were identified in a RIL population derived from a cross between Conrad (resistant) × Sloan (susceptible) parents [119]. These QTLs explained a small percentage of the phenotypic variance (3.6–9.2%) and were located on chromosomes 8, 13, 15, 16, and 19. These QTLs were not the same as those that confer resistance to *Phytophthora sojae*, suggesting that different loci are involved in resistance against these root rot pathogens [119]. Similar results were obtained from a genome-wide association study using cultivated and landraces of soybean [322]. This study identified 12 single nucleotide polymorphisms (SNPs) associated with *F. graminareum* resistance, which explained only a small percentage (5.53–14.71%) of the observed phenotypic variation.

A major QTL on chromosome 8 that explained 38.5% of the phenotypic variance was found in a RIL population derived from a cross between 'Wyandot' (partially resistant) × PI 567301B (highly resistant) [118]. This QTL harbored 39 genes, including the *Rhg4* locus for soybean cyst nematode (SCN) resistance [118]. Overall, nine genes coding for hydroxymethylglutaryl-CoA, a key enzyme in flavonoid biosynthesis pathway, were found in this QTL. In addition, there were three rapid alkanization factor (RALF) genes that can initiate a signal transduction pathway and two genes coding for subtilisin-like proteases. Subtilisin proteases are believed to be secreted into the extracellular matrix and function to reorganize cell wall components during defense response [118,323]. A subsequent study identified four differentially expressed genes that mapped to this QTL located on chromosome 8. These genes included an actin-related protein 2/3 complex subunit, an unknown protein, a hypothetical protein, and a chalcone synthase 3 [324]. This study demonstrated that removal of the seed coat of highly resistant soybean lines makes them susceptible to *F. graminareum*, indicating that proteins or secondary compounds in the seed coat may be involved in resistance [324].

### 4.1.6. Fusarium Root Rot in Cereals

The Fusarium species *F. avenaceum*, *F. graminareum*, *F. culmorum*, *F. verticillioides*, *F. pseudograminareum* are ubiquitous soil-borne fungus able to cause foot and root rot and Fusarium head blight or earmold on different small-grain cereals such as wheat, barley, maize, and oat [325–327]. The emphasis of this review is limited to horticultural crops. For a comprehensive review of Fusarium disease in cereals, the reader is directed to previous studies and reviews [104,328–335].

### 4.1.7. Phoma Root Rot

Studies related to the understanding of the molecular mechanism underlying Phoma resistance are scarce. A few studies were identified that focused on JA and thiabendazole's use to combat postharvest rots caused by *Phoma betae* and *P. sclerotioides* [336,337] or on the assessment of alfalfa cultivars for resistance against *P. sclerotioides* [156,338].

A recent study in onion utilized different sources of genetic resistance to *P. terrestris* to develop segregating families. One segregating family was scored for resistance and susceptibility with the resultant ratio of segregants fitting a single dominant locus. However, in another segregating family, the resulting segregation ratio did not fit a single dominant or a recessive locus [339]. The severity of root rot was mapped to a locus on chromosome 4, and it explained 28 to 35% of the phenotypic variation. Estimates of additive and dominance effects revealed that this source of resistance is co-dominantly inherited [339].

*P. terrestris* resistance was also assessed from a different source of resistance, and the resulting, co-dominantly inherited, QTL mapped to the same region on chromosome 4 and explained 54% of the phenotypic variation [339]. This study demonstrated that resistance from different genetic sources mapped to the same chromosome region and showed similar modes of inheritance [339].

### 4.1.8. *Thielaviopsis basicola* Root Rot

A QTL mapping study [340] and a proteomic analysis [143] are the only reports that provide some insight into the sources of resistance and defense response mechanisms

against *Thielaviopsis basicola* in cotton. Phenotypic variation between resistant and susceptible cotton lines was associated with three QTLs that explained 19.1, 10.3, and 8.5% of the total phenotypic variation [340]. This study provided a list of 624 candidate genes that were located within the identified QTL regions. The list included 22 pathogen defense and 36 stress-responsive genes. Fine mapping is required to narrow down this list of candidate genes for each QTL.

A time-course analysis of cotton root proteomes was performed during a compatible interaction with *T. basicola* [143]. Analysis of root extracts was conducted at 1, 3, 5, and 7 days post-inoculation. The study found that more plant proteins were down-regulated, especially in the early stages of infection, than upregulated. A total of 58 protein clusters were found to be upregulated across the time-course analysis. According to their putative biological role, these 58 protein clusters were further identified and classified into five major categories: defense, stress, primary and secondary metabolism, and diverse function. A number of the upregulated proteins corresponded to PR, with the majority of them belonging to the PR-10 family. The function of PR-10 genes is still unknown; however, the authors suggested that these proteins may be involved in hormone-mediated disease resistance in cotton [143]. A putative thaumatin protein, another PR protein, was also upregulated during *T. basicola* infection. The two additional pathogen defense proteins corresponded to a Meloidogyne-induced protein MIC-3, which were originally correlated with the disruption of nematode development in cotton [341]. The molecular function of the MIC-3 and the MIC family remains unknown, mainly due to the absence of known functional motifs and domains [342].

### 4.1.9. A Transgenic Approach to Counter *Thielaviopsis basicola* Root Rot

Transgenic cotton lines expressing AtNPR1 (nonexpresser of PR1) gene were found to be significantly tolerant to *T. basicola* [157]. The roots of the transgenic lines tended to recover faster after *T. basicola* infection. Transgenic plants also showed higher shoot and root mass, longer shoot length, and a greater number of boll-set than wild-type plants after *T. basicola* infection. NPR1 is a regulatory protein that participates as a critical positive regulator of the SA-dependent signaling pathway and systemic acquired resistance. Transcriptional analysis of transgenic roots exhibited stronger and faster induction of PR proteins such as PR1, thaumatin, glucanase, lipoxygenase (LOX1), and chitinase [157].

### *4.2. Oomycete Root Rot*
### 4.2.1. Aphanomyces Root Rot

*A. euteiches* cause high yield losses in legumes such as pea and alfalfa. It has been challenging to investigate the genetic basis of resistance in these two plant species due to their complex and partial genome information, the polygenic inheritance of resistance, and difficulties in field-based phenotyping. *Medicago truncatula,* with a much simpler genome, has been used as a surrogate model to understand the molecular interactions and resistance mechanism against *A. euteiches* [343]. The key molecular responses associated with *A. euteiches* resistance in *Medicago truncatula* are presented in Figure 3.

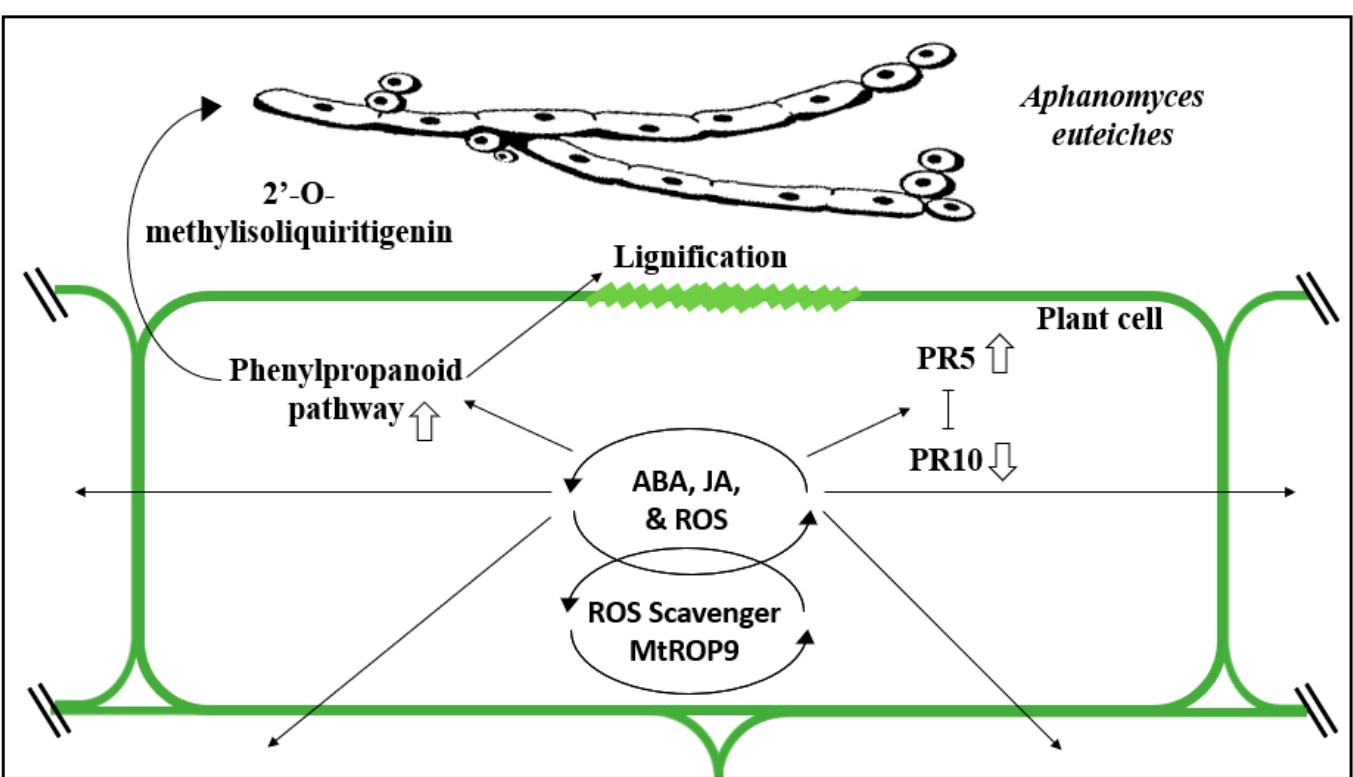

**Figure 3.** A model representing a resistance response following inoculation with *Apahanomyces euteiches* in *Medicago truncatula*. Events include the synthesis and signaling of abscisic acid (ABA) and reactive oxygen species (ROS). ROS are regulated by a small GTPase, named MtROP9. ABA and ROS signaling result in the overexpression of PR5 gene and the phenylpropanoid pathway. The lignification and synthesis of 2′-O-methylisoliquiritigenin are positively correlated with disease resistance against *A. euteiches*. 2′-O-methylisoliquiritigenin was shown to significantly impede *A. euteiches* development and zoospore germination. Model derived from results presented in Badis et al., (2015); Colditz et al., (2004); Djébali et al., (2009); Leonard M Kiirika et al., (2014); Leonard Muriithi Kiirika et al., (2012); Nyamsuren et al., (2003); Trapphoff et al., (2009) [344–350].

A monogenic control of resistance against *A. euteiches* has been reported in several studies. A QTL named AER1 was mapped to the distal part of chromosome 3 [346,351]. The genomic region corresponding to the QTL contains a supercluster of nucleotide-binding site leucine-rich repeat (NBS-LRR) genes [346,351]. This region also included proteasome-related genes, a cluster of nine F-box protein–encoding genes, and one gene coding for a ubiquitin-associated enzyme [352]. qRT-PCR data showed that the ubiquitin-associated enzyme and three F-box-encoding genes were induced in a resistant line (A17) following pathogen inoculation but not in the susceptible line. F-box proteins are known to be involved in hormone regulation and in plant immunity [353,354], regulation of pericycle cell divisions [355], and lateral root production [356].

The highly redundant presence of ABA-responsive proteins indicates that ABA-mediated signaling is involved in the interaction between *M. truncatula* and *A. euteiches* (Figure 3) [345,349]. ABA production and signaling are known to contribute to JA accumulation, as well as for the activation of resistance against *Pythium irregulare* in Arabidopsis [357,358]. Furthermore, PR10 family proteins, interspersed within the ABA-responsive genes, increased in *M. truncatula* after *A. euteiches* infection. However, a later study reported that the accumulation of PR10 protein was mainly correlated with *A. euteiches* proliferation and not plant resistance [352]. RNAi-mediated suppression of several PR10 genes led to a reduced *A. euteiches* colonization, which was linked with a parallel induction of a set of PR5 proteins (thaumatin-like proteins) (Figure 3) [359]. Therefore, PR10 and PR5 proteins act antagonistically.

A significant link has been found between enhanced synthesis and accumulation of flavonoid compounds and resistance against *A. euteiches* (Figure 3) [344]. Transcriptome and proteome analyses revealed a strong induction of chalcone synthase and isoflavone reductase genes after pathogen challenge [350,352,360,361]. Furthermore, the gene coding for isoliquiritigenin 2′-O-methyltransferase showed the highest induction after *A. euteiches* infection in the most resistant lines [345]. The metabolite 2′-O-methylisoliquiritigenin was shown to significantly impede *A. euteiches* development and zoospore germination (Figure 3) [344].

The accumulation of lignin has also been linked with partial resistance against *A. euteiches* [344,346]. Furthermore, the higher accumulation of lignin in resistant plants is associated with more efficient hydrogen peroxide ($H_2O_2$) scavenging mechanisms that are activated due to infection (Figure 3) [362]. ROS are activated following *A. euteiches* inoculation and are regulated by a small GTPase, named MtROP9 [347,348,350]. The knockdown of MtROP9 in *M. truncatula* resulted in three primary outcomes: (1-) prevented the detection of respiratory burst oxidase homologs, (2-) led to reduced activity of enzymes involved in the primary antioxidative processes; and (3-) promoted *A. euteiches* hyphal root colonization [347,348].

### 4.2.2. Role of Nodulation and Mycorrhizas in Aphanomyces Root Rot

The Nod Factor perception (NFP) gene involved in nodulation was reported to confer resistance against *A. euteiches* in *M. truncatula* [360]. NFP knockout mutants were significantly more susceptible to *A. euteiches* than wild-type, while NFP overexpressing lines showed increased resistance. Transcriptome analyses showed that knockout of the NFP gene led to changes in the expression of more than 500 genes involved in dynamic cell processes associated with disease response [360].

The NF-YA1 gene, a central transcriptional regulator of symbiotic nodule development, determined susceptibility to *A. euteiches* [363]. The Mtnf-ya1-1 mutant plants showed a better survival rate and reduced symptoms as compared to their wild-type background. Comparative analysis of the transcriptome of wild-type and Mtnf-ya1-1 mutant lines resulted in identifying 1509 differentially expressed genes. Among these differentially expressed genes, 36 defense-related genes were constitutively expressed in Mtnf-ya1-1, while 20 genes linked to hormonal, notably auxins, Ethylene and ABA, pathways were repressed [363].

Mycorrhiza seems to impart a bioprotective effect, observed earlier in pea roots against *A. euteiches* [364,365]. The increased resistance is postulated to be due to the following reasons: (I) enhanced physical resistance and damage compensation capability of the plant due to improved nutritional status, (II) changes in the microbial populations of the mycorrhizosphere, (III) competition between invading microorganisms (IV), increased production of secondary metabolites that have antimicrobial properties (V) activation of plant defense mechanism via accumulation of defense-related proteins [352,364]. Histochemical analysis of both microorganisms in the roots revealed a competition for physical space, which likely reduces *A. euteiches* hyphae or oospores, resulting in diminished disease symptoms [352].

### 4.2.3. Pythium Root Rot

The molecular mechanism of resistance to Pythium root rot has been primarily investigated in soybean and the common bean, however, recently, Pythium-responsive genes have been reported in apples as well.

In soybean, five QTLs associated with resistance to *P. sojae* were mapped to chromosomes 1, 6, 8, 11, and 13 [216]. Each QTL explained 7.9 to 17.8% of the phenotypic variation. QTLs associated with resistance to other root rot pathogens colocalize with the QTLs associated with *P. sojae* resistance. Chromosome 1 QTL colocalized with a QTL associated with resistance for *Phytophthora sojae* [366]. The chromosome 6 QTL was closely located to a QTL reported for *Phytophthora sojae* [367] and *Fusarium virguliforme* [368]. The QTL on chromosome 8 was found in a region associated with resistance for *F. virguliforme* [369]. The

QTL on chromosome 13 was located in an area associated with resistance to several other soybean pathogens, including *Phytophthora sojae* [370], *F. virguliforme* [371], and *F. graminareum* [119]. The QTLs on chromosomes 6 and 8 also colocalized with two QTLs found associated with resistance to *Pythium ultimum*. These QTLs, associated with *P. ultimum* resistance, on chromosomes 6 and 8 explain 7.5–13.5% and 6.3–16.8% of the phenotypic variance, respectively [372].

In common beans, the response of a set of 40 common genotypes to *P. ultimum* was investigated [373]. The emergence rate showed a significant association between seed coat color and response to this pathogen. In total, 11 bean genotypes with colored seeds exhibited a high percentage of emergence. A major gene (Py-1) controlling the emergence rate was mapped to the region of the gene P, an essential color gene involved in the control of seed coat color, located on linkage group (LG) 7. Using a RIL population of colored seeds, other two QTLs associated with the emergence rate and another with seedling vigor were identified on LG 3 and 6, respectively. QTL on LG6 was mapped to the gene V region, which is another gene involved in the genetic control of seed color.

The transcriptomic changes in apple root tissue when infected with *P. ultimum* were analyzed using tolerant and susceptible rootstock lines [374,375]. The mechanism of defense response involving the recognition of PAMPs, hormone signaling, and synthesis of PR genes was identified. Genes coding for proteins with predicted function of pathogen detection such as the chitin elicitor receptor kinase (CERK) and wall-associated receptor kinase (WAK) were among the differentially expressed genes identified in the resistant line. Genes associated with the biosynthesis and signaling of several phytohormones including Et, JA, and cytokinins were specifically induced in response to *P. ultimum* inoculation. The strong induction of cytokinin hydroxylase encoding genes suggests that cytokinin signaling may play a unique role in the defense response in apple roots. Furthermore, genes coding for secondary metabolism enzymes, cell wall fortification, PR proteins, laccase, mandelonitrile lyase, and cyanogenic beta-glucosidase were consistently up-regulated in the later stages of infection [374]. Like apple, in *Zingiber zerumbet* (shampoo ginger or wild ginger), high differential modulation of genes involved in cell wall fortification, lignin biosynthesis, and SA/JA hormone indicates that these genes play a central role in restricting *P. myriotylum* proliferation [376].

On a global scale, delayed or interrupted activation of multiple defense pathways seems to underlie susceptibility. This has been observed in various transcriptome analysis studies against root rots [304,305,308,362]. Similar observations were discernible from transcriptomic and microscopic data in the susceptible B.9 roots [32,375,377]. Microscopy data on the pathogen growth progress revealed a swift development of root necrosis in the most susceptible genotypes, with the entire root system becoming necrotic within a period of 24 h after initial infection [377]. The necrosis progression could be delayed for several days without the whole root tissues being engulfed for the most resistant genotypes.

### 4.2.4. Phytophthora Root Rot

Soybean is the species of choice to understand the molecular interactions between Phytophthora and the plant host. Phytophthora root rot (PRR) of soybean is the second leading cause of yield loss in soybean in North America, surpassed only by soybean cyst nematode [378].

More than 20 dominant genes, known as resistant to *P. sojae* (RPS) genes, associated with PRR resistance have been identified in the soybean genome, with most of them mapping to Chromosome 3 [201,367,368,379–383].

Once incorporated into soybean cultivars, Phytophthora race-specific resistance genes have a useful life of only 8 to 15 years before new virulent races of the pathogen evolve [201,380,384]. Intensive use of race-specific genetic resistance for control of *P. sojae* has resulted in the emergence of new races that are virulent to the current resistance genes [385]. Over 50 races of *P. sojae* have been reported in the literature [386,387]. Currently, none of the single host-resistance genes can counter all *P. sojae* races. Several reports have

highlighted the importance of marker-assisted selection (MAS) to pyramid several QTLs in soybean cultivars to reduce losses by PRR [367,368,379,380,388]. This approach would help in reducing the selection pressure for new virulent races of *P. sojae*. Partial resistance or tolerance, also called quantitative disease resistance, generates a lower selection pressure on the pathogen population; therefore, it is expected that partial resistance will be more durable than general race-specific resistance.

The Rps1k gene in soybean has garnered significant interest because it confers stable resistance to broad-spectrum *P. sojae* strains in the USA. The Rps1k gene locus, cloned as part of a bacterial artificial chromosome (BAC), carries two classes of coiled coil-nucleotide binding-leucine rich repeat ((CC)-NBS-LRR) genes [389,390]. *Rps2* and *Rps4* gene loci were cloned, and they were also characterized as NBS-LRR genes [382,391]. In RpsJS gene locus, 14 predicted genes exist, with three being NBS-LRR type genes [392]. The RpsYD29 gene was mapped to a region with two NBS-LRR type genes [393]. These two genes showed high similarity to the NBS-LRR present in the Rps1k gene locus. *Rps10* gene has been mapped to chromosome 17, where eight putative candidate genes were found. In total, two candidate genes encoding serine/threonine (Ser/Thr) protein kinases were identified [394]. The identity and function of the remaining RPS genes remain unknown.

In soybean roots, a strong correlation between the extent of preformed suberin in soybean roots and the resistance to *P. sojae* was observed [395]. As a cell wall component, suberin is known to constitute a barrier to the pathogen and also acts as a toxin to microbes due to its high concentration of phenolic compounds [395,396]. To colonize the root, *P sojae* hyphae grow through the suberized middle lamellae between epidermal cells. This process took 2 to 3 h longer in Conrad (resistant genotype) than in OX760-6 (partially resistant genotype) [384]. Subsequent growth of hyphae through the endodermis was also delayed in Conrad. The delay in the progression of *P. sajoe* in the resistant cultivar provides this genotype with more time to activate and establish its chemical defenses. Additionally, Conrad had more preformed aliphatic suberin and was induced to form more aliphatic suberin after initial infection than OX760-6. The authors concluded that suberin's synthesis provides a target for the selection and development of new soybean cultivars with higher levels of partial resistance to *P. sojae*.

Expression of a number of micro RNAs (miRNAs) was found significantly altered upon infection with *P. sojae* in resistant and susceptible genotypes [397]. Further analyses revealed many reciprocally inverse patterns of the miRNA-gene target pairs upon infection. These expression patterns propose a feedback circuit between miRNAs and protein-coding genes. A knock down of miRNA 393 led to enhanced susceptibility of soybean to *P. sojae*, as well as to a reduction in the expression of isoflavonoid biosynthetic genes [398]. On the other hand, overexpression of miRNA gma-miR1510a/b in the hairy roots of soybean resulted in enhanced susceptibility to *P. sojae* [399]. Results showed that miR1510 guides the cleavage of the *Glyma.16G135500* gene, which encodes an NBS-LRR gene. These results suggest a pivotal role of both miRNAs in resistance against *P. sojae*. As illustrated by this example, the role of miRNA in plant defense needs to be investigated broadly in other plants.

### 4.2.5. Phytophthora Root Rot in Other Crops

*P. nicotianae* is a major problem in tobacco production. At least six QTLs were mapped in tobacco that contribute to high level of resistance against *P. nicotianae* [400]. All six QTLs explained 64.3% of the phenotypic variation, while the two largest QTLs explained 25.4 and 20.4% of the observed phenotypic variation. The major QTL on linkage group four was found to co-segregate with *Abl*, a gene involved in accumulation of *cis*-abienol [400]. This compound is exuded by trichomes and has been previously associated with roles in plant defense against insects, plant pathogens, and other microbes [400–402]. Recent studies have identified resistant and susceptible genes using RNA-seq time-course analyses [403,404]. Some resistance gene candidates include disease-resistance proteins, chiti-

nases, pathogenesis-related proteins, calcium-dependent and -binding proteins, mitogen-activated protein kinases, transcription factors, among others.

*P. rubiis* is one of the most serious and destructive diseases of raspberry [405,406]. The two major QTLs, located on LG 3 and 6, associated with Phytophthora resistance, have been identified [406]. Root vigor and disease resistance mapped to the same major QTL on LG 3. An auxin receptor or germin-like protein mapped to this LG 3 QTL. This QTL is possibly involved in the initiation of new axes of growth as a defense response. The effect of the LG 6 QTL has only been identified at the infected site. Therefore, LG 6 QTL may be better interpreted as a resistance locus rather than a vigor-related gene [406].

In avocado, the transcript levels of defense-related genes were characterized and compared among five rootstocks with varying resistance to root rot [407]. The results indicated the involvement of PR-5 and endochitinase in the defense response. However, neither of the genes could be directly linked to the observed resistance. The difference in transcript abundance of phenylalanine ammonia-lyase and lipoxygenase genes was also observed when comparing resistant and less resistant rootstocks, indicating their potential involvement in the resistance.

In strawberry, five genes for resistance to thirty races of *P. fragariae* have been identified, including Rpf1 that was characterized as a dominant monogenic gene that confers resistance to at least 18 races of *P. fragariae* [197,408]. However, none of the five genes have been characterized or associated with any known defense mechanism.

## 5. Conclusions

Moderate to high levels of resistance have been identified in breeding lines or cultivars of most crops affected by root rots. Furthermore, significant progress has been made in identifying genes that respond to fungi and oomycete root rot pathogens. It is evident that no universal response controls the resistance against this heterogeneous group of pathogens. The reactions are highly dependent on the host genetics and the pathogen involved. Resistant responses are governed, mostly, by multiple mechanisms and genes and, on rare occasions, by a single, independent, dominant gene. Hormones that drive responses against root rots vary for each individual host-pathogen interaction. The role of SA, JA, Et, ABA, cytokinins, or any other hormone, should be studied on an individual basis.

Hundreds of studies have opted for the screening of resistance QTLs. In some of these studies, a major resistance QTL was identified. However, for most of these QTLs, the genes underlying resistance remain to be elucidated. The identification of these genes might help confer broad resistance against root rots. The host's rapid response seems to be a shared feature that determines resistance against root rot pathogens. Resistant and susceptible genotypes commonly respond with the same set of defense genes, however, the slower speed of response in the susceptible genotypes results in pathogenesis. This standard feature was documented in the rice-*R. solani*, potato-*R. solani*, pea-*F. solani*, *M. truncatula*-*A. euteiches*, and apple-*Pythium ultimum* interactions. Interestingly, legumes are predominant hosts of root rot pathogens. A shared factor among legumes might explain this susceptibility. A vital factor to consider is that legumes are the only known plant taxon that forms a symbiotic relationship with Rhizobium spp. Symbiosis-related mechanisms might be a gateway highjacked by root rot pathogens.

Emerging high-throughput phenotyping technologies will allow for efficient detection of root rot resistant lines across the entire array of plant hosts. Combined with field-scale phenotyping, high-throughput genotyping and genomics approaches are expected to help in identifying genes involved in pathogen resistance or susceptibility. This information can be utilized in breeding or genome editing approaches to develop resistant crops that can be cultivated sustainably.

**Author Contributions:** B.A.W.-B. and A.D. wrote, have read, and agreed to the published version of the manuscript. All authors have read and agreed to the published version of the manuscript.

**Funding:** B.A.W.-B. acknowledges graduate research assistantship support from Washington State University Graduate School. Work in the Dhingra lab in the area of crop improvement is supported in part by Washington State University Agriculture Research Center Hatch grant WNP00011, and pea improvement by USA Dry Pea and Lentil Council.

**Institutional Review Board Statement:** Not applicable.

**Informed Consent Statement:** Not applicable.

**Conflicts of Interest:** The authors declare no conflict of interest.

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
