# Peer review of "Understanding Root Rot Disease in Agricultural Crops"

_horticulturae, doi:10.3390/horticulturae7020033_

Round 1
Reviewer 1 Report
an impressive review of the examined sector with a huge references list.
no remarks and no mistakes to report.
only few references with authors name in capital letters but i don't know if it is allowed by the review rules.
a personal comment: i would have liked some more news about interaction between resistance mechanism and other potential control measures as soil bioma.
Author Response
- An impressive review of the examined sector with a huge references list.
No remarks and no mistakes to report.
Author’s Response: Thank you for your supportive comments.
- Only few references with authors name in capital letters but i don't know if it is allowed by the review rules.
Author’s response: We have addressed this issue.
- A personal comment: I would have liked some more news about interaction between resistance mechanism and other potential control measures as soil bioma.
Author’s Response: Thank you for the interesting suggestion. The authors opine that such a topic would be out of the scope of the current focus on the review manuscript.
Reviewer 2 Report
The purpose of this review was to provide a very comprehensive overview of the crops affected by root rot, the causative agents, and what we know so far about the genetics of resistance. Clearly a tremendous effort went into a literature search to create the manuscript. It’s a very extensive paper! However, what is lacking are themes or a story of what we have learned from this body of work. Some suggestions are below
Main concerns
- Are there patterns regarding where resistant cultivars originate vs centers of origin of the crop species or centers of origin of the rot causative agent? For example, is there evidence that crops grown outside the region in which they were domesticated (such as potatoes grown outside of South America) were then subject to root rot that was not such a major issue in their center of origin? Could examining such patterns reveal genetic resources for obtaining resistance (meaning, would there be local cultivars or wild relatives that could be resistant?)? Or, are there temporal patterns to when rot pathogens became problematic for different crops?
- Are there patterns or categories of resistance genes (either patterns between related crop species, or patterns of what is effective for different types of rot causative organisms)? Perhaps it would help to map/identity known resistance factors on the Figures. While many of the QTLs have yet to be identified, many have a primary gene of influence.
- Can this work be tied into the challenges of crop breeding? It seems worth commenting on the issue of moving over several QTLs from one cultivar to another. Also, many of our agriculture crops have a narrow gene pool, but wild relatives do hold great genetic potential for improvement. Improvement by GMO methods (the definition for which is different and subject to vastly different regulations across the globe) is promising alternative and one that does merit more discussion. The challenge of moving from lab to greenhouse to field testing to agricultural usage is considerable.
- In addition to the intrinsic lack of resistance and factors listed in lines 42 onward, is there evidence of root rot causative agents remaining in environment reservoirs such as residual material left post harvest or in nearly wild plants? Or is there evidence for transfer of infective agents via shared field equipment or contaminated footwear?
- The broad host range of some of the pathogens seems a possible theme to expand on and tie together some of the work. For example, Rhizoctonia solani, which (as mentioned in lines 303-305) can infect monocots (rice) and dicots (beets), illustrates that many of these pathogens can be generalists. Such organisms may be challenging to control as they can clearly evade host immune systems from a wide variety of species. Having resistant rice will only help that crop, leaving other species (like beets) still susceptible to infection.
Minor concerns
- Are there themes about which pathogens are opportunistic? Which ones need live tissues?
- Please consider standardizing table formats and contents. For example, Table 1 does not report host range or distribution, which is reported in subsequent tables. While Table 1 lists “host” in alphabetical order by common name, Table 2 lists “plants attacked” in no particular order.
- Please clarify whether resistance observed or under development is due to selective breeding or to genetic modification techniques (eg. Lines 178-180).
- Was permission obtained for adapting figure 1, figure 2, and figure 3? Also, these figures show the plant cells and pathogen cells existing side by side. It would be informative to show more of the interaction (either physical or molecular) between the organisms.
- It would be helpful to list the category (fungal, oomycete, bacterial) of root rot causative agents in lines 32-35 as readers may not be immediately familiar with all these examples.
- Please add a citation to line 52 for species survival for 10 years in soil.
- It is suggested to change line 53-54 from “therefore, crop rotation is not a practical strategy” to “therefore, crop rotation may not be fully effective as a control method.”
- The extensive reported symptoms in Table 4 makes for a very long table, can it be summarized into disease name as in prior tables or moved to supplementary?
- Please clarify all terminology in tables, such as the distinction between “worldwide” and “cosmopolitan.” Also, sometimes the continent is listed by itself, and sometimes countries are listed after the continent. Please standardize the location format.
- It would be worth noting that Phytophthora causes extensive potato tuber damage as well as symptoms to aerial portions (lines 208 onward).
- Cassava is a staple crop in some countries and brown streak is a serious issue for this crop. Please consider expanding this section as there is a great deal more information than was briefly summarized. For example, efforts have been underway for years to develop resistant transgenic casava.
17. The introductory paragraph needs additional citations for the crop losses and legumes as a common host.
18. The conclusions would benefit from ending with some future directions. Where do we go from here?
Author Response
Reviewer 2:
The purpose of this review was to provide a very comprehensive overview of the crops affected by root rot, the causative agents, and what we know so far about the genetics of resistance. Clearly a tremendous effort went into a literature search to create the manuscript. It’s a very extensive paper! However, what is lacking are themes or a story of what we have learned from this body of work. Some suggestions are below
Main concerns
- Are there patterns regarding where resistant cultivars originate vs centers of origin of the crop species or centers of origin of the rot causative agent? For example, is there evidence that crops grown outside the region in which they were domesticated (such as potatoes grown outside of South America) were then subject to root rot that was not such a major issue in their center of origin? Could examining such patterns reveal genetic resources for obtaining resistance (meaning, would there be local cultivars or wild relatives that could be resistant?)? Or, are there temporal patterns to when rot pathogens became problematic for different crops?
Author’s response: The reviewer raises an interesting question. In our literature review, we did not come across historical documentation of how and when rot pathogens became pathogenic, or if there are locally adapted cultivars. While we could not include any updates from the suggested perspective, it is an interesting topic nevertheless.
- Are there patterns or categories of resistance genes (either patterns between related crop species, or patterns of what is effective for different types of rot causative organisms)? Perhaps it would help to map/identity known resistance factors on the Figures. While many of the QTLs have yet to be identified, many have a primary gene of influence.
Author’s response: As is evident from the studies covered in the review the responses are highly variable depending on the host genetics and the pathogen involved. The only and speculative noted pattern may be the speed of response of the host, which was addressed in the conclusion section.
- Can this work be tied into the challenges of crop breeding? It seems worth commenting on the issue of moving over several QTLs from one cultivar to another. Also, many of our agriculture crops have a narrow gene pool, but wild relatives do hold great genetic potential for improvement. Improvement by GMO methods (the definition for which is different and subject to vastly different regulations across the globe) is promising alternative and one that does merit more discussion. The challenge of moving from lab to greenhouse to field testing to agricultural usage is considerable.
Author’s response: This is another great suggestion. However, it is outside the scope of this review.
- In addition to the intrinsic lack of resistance and factors listed in lines 42 onward, is there evidence of root rot causative agents remaining in environment reservoirs such as residual material left post harvest or in nearly wild plants? Or is there evidence for transfer of infective agents via shared field equipment or contaminated footwear?
Author’s response: Indeed, there is evidence of root rot causative agents remaining in “environmental reservoirs” that can survive up to 10 years (please refer to line 52). Evidence of postharvest-associated root rot is also mentioned on lines 208, 252, 271, and 566, as well as tables 2 and 4. The possibility of transfer of infective agents via shared field equipment was addressed in lines 56-57.
- The broad host range of some of the pathogens seems a possible theme to expand on and tie together some of the work. For example, Rhizoctonia solani, which (as mentioned in lines 303-305) can infect monocots (rice) and dicots (beets), illustrates that many of these pathogens can be generalists. Such organisms may be challenging to control as they can clearly evade host immune systems from a wide variety of species. Having resistant rice will only help that crop, leaving other species (like beets) still susceptible to infection.
Author’s response: Great point. This issue is addressed in lines 49-57.
Minor concerns
- Are there themes about which pathogens are opportunistic? Which ones need live tissues?
Author’s response: Most of the pathogens mentioned in this review are considered necrotrophs (live from dead tissue). However; there are a few exceptions that are characterized as hemi-biotrophs, such as some anastomosis groups of Rhizoctonia solani. While the nature of the pathogen is an important factor in the disease triangle, the focus of this review is to address the resistance/susceptibility of hosts.
- Please consider standardizing table formats and contents. For example, Table 1 does not report host range or distribution, which is reported in subsequent tables. While Table 1 lists “host” in alphabetical order by common name, Table 2 lists “plants attacked” in no particular order.
Author’s response: The tables were formatted differently, in some cases, to provide relevant information for the readers. Formatting also changes based on the availability of data. Table 1 is formatted differently from the other three tables because Rhizoctonia solani had and has been categorized into anastomosis groups based on what crop they affect. The category “plants attacked” in tables 2, 3, and 4 were organized alphabetically as suggested.
- Please clarify whether resistance observed or under development is due to selective breeding or to genetic modification techniques (eg. Lines 178-180).
Author’s response: Clarified as suggested (Line 183).
- Was permission obtained for adapting figure 1, figure 2, and figure 3? Also, these figures show the plant cells and pathogen cells existing side by side. It would be informative to show more of the interaction (either physical or molecular) between the organisms.
Author’s response: Figures 1, 2, and 3 represent original illustrations developed based on the findings in the cited articles. The information included in the figures/models included only validated genes/mechanisms known to be part of the resistance/susceptibility process. Many more genes could be added to the models; however, their relationship to resistance/susceptibility remains unclear at this time.
- It would be helpful to list the category (fungal, oomycete, bacterial) of root rot causative agents in lines 32-35 as readers may not be immediately familiar with all these examples.
Author’s response: This suggestion was addressed (lines 32-35).
- Please add a citation to line 52 for species survival for 10 years in soil.
Author’s response: Done (line 52)
- It is suggested to change line 53-54 from “therefore, crop rotation is not a practical strategy” to “therefore, crop rotation may not be fully effective as a control method.”
Author’s response: Done (lines 53-54)
- The extensive reported symptoms in Table 4 makes for a very long table, can it be summarized into disease name as in prior tables or moved to supplementary?
Author’s response: This suggestion was addressed. Reported symptoms were summarized where possible (in more than 50% of the original content).
- Please clarify all terminology in tables, such as the distinction between “worldwide” and “cosmopolitan.” Also, sometimes the continent is listed by itself, and sometimes countries are listed after the continent. Please standardize the location format.
Author’s response: The word cosmopolitan was replaced for worldwide when cosmopolitan was not defined within the cells of the table. When the continent was listed, a more detailed description was added such as - All areas cultivated with alfalfa in North America and Australia.
- It would be worth noting that Phytophthora causes extensive potato tuber damage as well as symptoms to aerial portions (lines 208 onward).
Author’s response: While some Phytophthora spp attack potatoes, the Phytophthora spp covered in this review produce diseases other than root rot. A sentence to include the symptoms caused in potato has been included.
- Cassava is a staple crop in some countries and brown streak is a serious issue for this crop. Please consider expanding this section as there is a great deal more information than was briefly summarized. For example, efforts have been underway for years to develop resistant transgenic casava.
Author’s response: We addressed this suggestion (lines 273-282)
- The introductory paragraph needs additional citations for the crop losses and legumes as a common host.
Author’s response: We added more citations as requested by the reviewer (lines 24-27)
- The conclusions would benefit from ending with some future directions. Where do we go from here?
Author’s response: As suggested, additional information has been included in the conclusions section (lines 871-877).
Reviewer 3 Report
In this review, the authors have collected a lot of root rot related information from aspects of pathogens, and crop hosts which are of great value for the research community. This MS is data-driven and was well prepared except for some issues to need to be addressed before publication.
Resistance information about tobacco to root rot should be provided, especially for Phytophthora nicotianae.
The recent studies of small RNAs and their role in plant root rot disease should be included in this MS.
Author Response
Reviewer 3:
In this review, the authors have collected a lot of root rot related information from aspects of pathogens, and crop hosts which are of great value for the research community. This MS is data-driven and was well prepared except for some issues to need to be addressed before publication.
- Resistance information about tobacco to root rot should be provided, especially for Phytophthora nicotianae.
Author’s response: Information about tobacco root rot had already been included (see lines 129 and 314, 361, Tables 3 and 4). We also expanded on Phytophthora nicotianae root rot in tobacco (lines 817-827).
- The recent studies of small RNAs and their role in plant root rot disease should be included in this MS.
Author’s response: This is a great suggestion. Most of the small RNA studies focused on the interaction between soy and Phytophthora sojae. We included a section to address this topic (lines 801-812).
Round 2
Reviewer 2 Report
Thank you for the revisions to your manuscript. It is an enormous volume of work. This reader finds that overall themes/stories/take away messages are lacking, which were suggested additions in the initial review. However, that can be a writing style preference.
One concern still present is that some phrases and sentences were copies of those from the citations. Please carefully check for originality and make any needed revisions.
Author Response
We thank the reviewer for their feedback.
Thank you for the revisions to your manuscript. It is an enormous volume of work. This reader finds that overall themes/stories/take away messages are lacking, which were suggested additions in the initial review. However, that can be a writing style preference.
Author Response: We appreciate the suggestion by the reviewer. Their previous feedback was very useful and has improved the manuscript tremendously. As they have rightly pointed out, there is perhaps a writing style preference. We hope that the review is in a format where it is technically sound for publication.
One concern still present is that some phrases and sentences were copies of those from the citations. Please carefully check for originality and make any needed revisions.
Author Response: The manuscript was processed through iThenticate, a plagiarism detection software. The only significant similarity was detected with the preprint of this article posted on preprints.org.